# Open-World Object Manipulation using Pre-Trained Vision-Language Models

**Austin Stone**[*]**, Ted Xiao**[*]**, Yao Lu**[*]**, Keerthana Gopalakrishnan,**
**Kuang-Huei Lee, Quan Vuong, Paul Wohlhart, Sean Kirmani,**
**Brianna Zitkovich, Fei Xia, Chelsea Finn, Karol Hausman**
Robotics at Google

**Abstract:** For robots to follow instructions from people, they must be able to connect the rich semantic information in human vocabulary, e.g. "can you get me the pink stuffed whale?" to their sensory observations and actions. This brings up a notably difficult challenge for robots: while robot learning approaches allow robots to learn many different behaviors from first-hand experience, it is impractical for robots to have first-hand experiences that span all of this semantic information. We would like a robot's policy to be able to perceive and pick up the pink stuffed whale, even if it has never seen any data interacting with a stuffed whale before. Fortunately, static data on the internet has vast semantic information, and this information is captured in pre-trained vision-language models. In this paper, we study whether we can interface robot policies with these pre-trained models, with the aim of allowing robots to complete instructions involving object categories that the robot has never seen first-hand. We develop a simple approach, which we call Manipulation of Open-World Objects (MOO), which leverages a pre-trained vision-language model to extract object-identifying information from the language command and image, and conditions the robot policy on the current image, the instruction, and the extracted object information. In a variety of experiments on a real mobile manipulator, we find that MOO generalizes zero-shot to a wide range of novel object categories and environments. In addition, we show how MOO generalizes to other, non-language-based input modalities to specify the object of interest such as finger pointing, and how it can be further extended to enable open-world navigation and manipulation. The project's website and evaluation videos can be found at `https://robot-moo.github.io/`.

## 1   Introduction

For a robot to be able to follow instructions from humans, it must cope with the vast variety of language vocabulary, much of which may refer to objects that the robot has never interacted with first-hand. For example, consider the scenario where a robot has never seen or interacted with a plush animal from its own camera, and it is asked, "can you get me the pink stuffed whale?" How can the robot complete the task? While the robot has never interacted with this object category before, the internet and other data sources cover a much wider set of objects and object attributes than the robot has encountered in its own first-hand experience. In this paper, we study whether robots can tap into the rich semantic knowledge captured in such static datasets, in combination with the robot's own experience, to be able to complete manipulation tasks involving novel object categories.

Computer vision models can capture the rich semantic information contained in static datasets. Indeed, composing modules for perception, planning, and control in robotics pipelines is a long-standing approach [1, 2, 3], allowing robots to perform tasks with a wide set of objects [4]. However, these pipelines are notably brittle, since the success of latter motor control modules relies on precise object localization. On the other hand, several prior works have trained neural network policies with pre-trained image representations [5, 6, 7, 8] and pre-trained language instruction embeddings [9, 10, 11, 12]. While this form of vanilla pre-training can improve efficiency and generalization, it does not provide a mechanism for robots to ground and manipulate novel semantic concepts, such as unseen object categories referenced in the language instruction. This leads to a crossroads — some approaches can conceivably generalize to many object categories but rely on fragile pipelines; others are less brittle but cannot generalize to new semantic object categories.

---

[*]Indicates equal contribution. Please direct correspondence to `tedxiao@google.com`.

7th Conference on Robot Learning (CoRL 2023), Atlanta, USA.

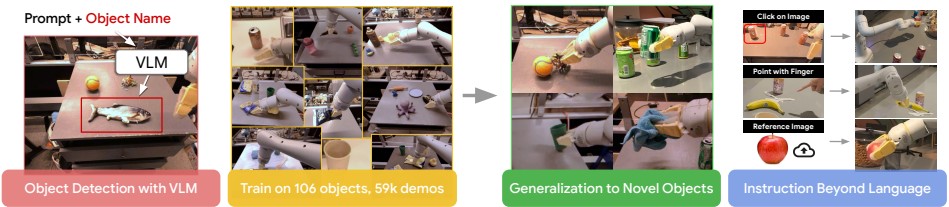

Figure 1: Overview of MOO. We train a language-conditioned policy conditioned on object locations from a frozen VLM. The policy is trained on demonstrations spanning a set of 106 objects using VLM-based object-centric representations, enabling generalization to novel objects, locations produced from new modalities.

To allow robots to generalize to new semantic concepts, we specifically choose to leverage open-vocabulary pre-trained vision-language models (VLMs), rather than models pre-trained on one modality alone. Such models capture the rich information contained in diverse static datasets, while grounding the linguistic concepts into a perceptual representation that can be connected to the robot's observations. Crucially, rather than using the pre-trained model for precise state estimation in its entirety (akin to pipelined approaches), we only use the VLM to identify the relevant objects in the image by coarsely localizing them, while allowing an end-to-end trained policy to use this information along with the original observation to perform the task. More specifically, our system receives a language instruction and uses a VLM to identify the 2D image coordinates of objects in the instruction. Along with the image and the instruction, the 2D coordinates of the objects are fed into our manipulation policy allowing it to ground the natural language to objects and know which objects to act upon without seeing any demonstrations with those objects. The VLM is frozen throughout all of our training, and the policy is trained with the real VLM detector in the loop to prevent the brittleness that can plague prior pipelined approaches.

The main contribution of this paper is a flexible approach for open-world object manipulation that interfaces policy learning with pre-trained vision-language models. An overview is given in Fig. 1. The pre-trained models are trained on massive static image and language data that far exceeds the robot's own experience. The robot's policy is trained to perform skills using demonstration data covering a more modest yet still physically diverse set of 106 training objects. The composition of the pre-trained vision-language model and the control policy leads to an overarching language-conditioned policy that can complete commands that refer to novel semantic categories.

We study the performance of our method across $1,472$ evaluations on a real robotic manipulator, where we find that our approach is significantly more successful than recent robot learning methods. Beyond verbal object descriptions, we also find that the trained policy can be easily combined with other means of communicating intent, e.g., pointing at an object and inferring the object description using a VLM, showing a generic image of the object of interest, or using a simple GUI. Finally, our experiments further show that our method can be integrated with an open-vocabulary object navigation model called Clip-on-Wheels (CoW), to complete mobile manipulation tasks involving novel objects. Throughout this paper, we refer to our approach as Manipulation of Open-vocabulary Objects (MOO) and the integrated mobile manipulation system as CoW-MOO.

## 2 Related Work

**Leveraging Pre-Trained Models in Robotic Learning.** Using off-the-shelf vision, speech, or language models is a long-standing approach in robotics [13, 14, 10]. Modern pre-trained vision and language models have improved substantially over older models, and have played an increasing role in robotics research. A large body of prior work has trained policies on top of frozen or fine-tuned visual representations [5, 15, 6, 16, 17, 18, 19, 7, 8, 20, 21], while other works have leveraged pre-trained language models [22, 23, 9, 10, 11, 24, 25, 12]. Unlike these prior works, we aim to leverage vision-language models that ground language in visual observations. Our use of vision-language models enables generalization to novel semantic object categories, which cannot be achieved by using vision models or language models individually.

**Generalization in Robotic Learning.** A number of recent works have studied how robots can complete novel language instructions [26, 22, 23, 9, 10, 11, 27, 28, 24], typically focusing on instructions with novel combinations of words, i.e. compositional generalization, or instructions with novel ways

to describe previously-seen objects and behaviors. Our work focuses on how robots can complete instructions with entirely new words that refer to objects that were not seen in the robot's demonstration dataset. Other research has studied how robot behaviors like grasping and pick-and-place can be applied to unseen objects [29, 30, 31, 32, 33, 34, 35, 36, 37], focusing on generalization to visual or physical attributes. Our experimental settings require visual and physical object generalization but also require semantic object generalization. That is, unlike these prior works, the robot must be able to ground a description of a previously-unseen object category.

**Vision-Language Models for Robotic Manipulation.** Two closest related works to our approach are CLIPort [38] and PerAct [12] that use the CLIP vision-language model as a backbone of their policy. Both of these approaches have demonstrated impressive level of generalization to unseen semantic categories and attributes. Inspired by these works, we aim to expand them to more general manipulation settings by: i) removing the need for depth cameras or camera calibration, ii) expanding and demonstrating that the hereby introduced representation works with other modalities such as pointing to the object of interest, iii) moving beyond 2D manipulation tasks, e.g. demonstrating the approach on tasks such as reorienting objects upright as well as mobile manipulation tasks.

**Open-World Object Detection in Computer Vision.** Historically, object-detection methods have been restricted to a fixed category map covering a limited set of objects [39, 40, 41, 42]. These methods work well for the object categories on which they are trained, but have no knowledge of objects outside their limited vocabulary. Recently, a new wave of object detectors have emerged that aim to go beyond simple closed-vocabulary tasks by replacing the fixed one-hot category prediction with a shared image-language embedding space that can be used to answer open-vocabulary object queries [43, 44, 45, 46]. Typically these methods rely on internet-scale data in the form of pairs of image and associated descriptive text to learn the grounding of natural language to objects. Various methods have been employed to then extract object localization information in the form of bounding boxes and segmentation masks. In our work, we use the OWL-ViT detector due to it's strong performance in the wild and publicly available implementation [43].

## 3 Manipulation of Open-World Objects (MOO)

The key goal of MOO is to develop a policy that can leverage the visually-grounded semantic information captured by pre-trained vision-language models for generalization to object types not in the policy training set. More specifically, we aim to use the VLM to localize objects described in a given instruction, while allowing the policy to complete the task using both the instruction and the object localization information from the VLM. MOO accomplishes this in two stages. First, the current observation and the words in the instruction corresponding to object(s) are passed to the VLM to localize the objects. Then, the object localization information and the instruction sans object information are passed to the policy, along with the original observation, to predict actions.

The key design choice of MOO lies in how to represent object information encoded in VLMs and how to feed that information to the instruction-conditioned policy. In the remainder of this section, we first overview the set-up, then describe the design of these crucial aspects of the method, and finally provide an overview of the model architecture and the training procedure as well as describe practical implementation details that allows us to deploy MOO on real robots.

### 3.1 Problem Set-Up

Formally, we assume that the robot, with image observations $o \in \mathcal{O}$ and actions $a \in \mathcal{A}$, is provided with a set of expert demonstrations $\mathcal{D}_{\text{robot}}$ collected via teleoperation. Each demonstration corresponds to a sequence of observation-action pairs $\{(o_j, a_j)\}_{j=1}^T$ collected over a time horizon $T$, and is annotated with a structured language instruction $\ell$ for the task being performed in the demonstration. To help facilitate object generalization, we assume that these language instructions are structured as a combination of a template and a list of object descriptions within that template. For example, for the instruction $\ell =$ *"move yellow banana near cup,"* the template is *"move X near Y,"* and the object descriptions are $X =$ *"yellow banana"* and $Y =$ *"cup."* Inspired by RT-1 [24], in this work, we focus on five different types of skills defining the templates: *"pick X,"* *"move X near Y,"* *"knock X over,"* *"place X upright,"* and *"place X into Y,".*

All of the objects seen in the demonstrations are drawn from a set $\mathcal{S}_{\text{robot}}$, and our objective is to complete new structured language instructions with a seen template but novel objects that are not in

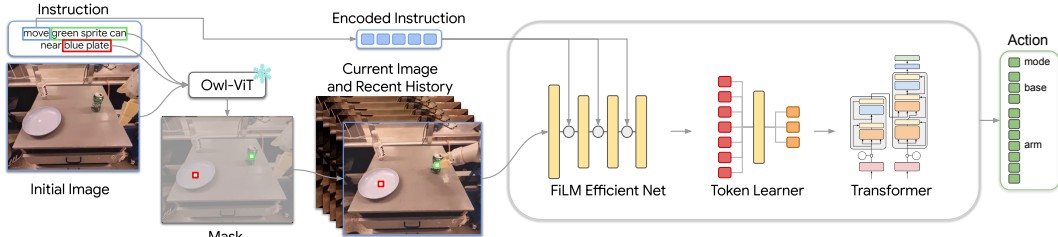

Figure 2: MOO architecture: We extract object location (represented as the center of the bounding box) on the first frame of an episode. The segmentation mask is concatenated channel-wise to the input image for each frame. We remove the language embedding for everything except the task so that the object specific information is only provided through the object instance mask.

$\mathcal{S}_{\text{robot}}$, which also have novel object descriptions. In aiming to complete this goal, our approach will leverage imitation learning and vision-language models, which we briefly review in the Appendix.

## 3.2 Representing Object Information

To utilize the object knowledge encoded in the VLMs, we need to pick a representation that can be easily transferred to the text-conditioned control policy. We start by identifying the instruction template (represented by verb $v$) and object $X$ (or list of objects $X, Y, ...$) from the instruction $\ell$. Equipped with an object description $X$, we query a VLM to produce a bounding box of the object of interest with the prompt $q = $ "An image of an $X$", and use the resulting detection (if any) as conditioning of our policy. To reduce the reliance of the exact segmentation of the object dimensions, we select a single pixel that is at the center of the predicted bounding box as the object representation. In the case of one object description, we use a single-channel object mask with the value set to $1.0$ at the pixel of the object's predicted location. In the case of two object descriptions, we set the pixel value of the first to be $1.0$ and the second to be $0.5$.

This design has two main advantages: first, it is a generic representation that works with objects of any size as long as they are visible, and second, it is compatible with a large selection of vision methods such as bounding boxes or segmentation masks as these can be easily transformed into a single, object-centric pixel location. We ablate other object representation choices in the experiments.

Importantly, this approach can handle object descriptions that are not previously seen in the robot's demonstration data, as long as it is sufficiently represented in the static large-scale training data of the VLM. For any unseen objects, we simply include a description in the task command, e.g., "pick *stuffed toy whale*." Once the object description is translated into a pixel location by the VLM, the robot's policy trained on demonstration data only needs to be capable of interpreting the mask location and how to physically manipulate the novel object's shape, rather than needing to also ground the semantic object description.

## 3.3 Architecture and Training of MOO

We present the model architecture and information flow of MOO in Fig. 2. As described above, we extract the object descriptions from the language instruction and together with the initial image feed them into the VLM to output object locations in the image. This information is then represented as an object mask with dots at the center of the objects of interest.

Once we obtain the mask, we append it channel-wise to the current image together with the recent image history, which is passed into the RT-1 policy architecture [24]. We use a language model to encode the previously extracted verb $v$ part of the language instruction in an embedding space of an LLM. The images are processed by an EfficientNet [47] conditioned on the text embedding via FiLM [48]. This is followed by a Token Learner [49] to compute a small set of tokens, and finally a Transformer [50] to attend over these tokens and produce discretized action tokens. We refer the reader to the RT-1 paper for details regarding the later part of the architecture. The action space corresponds to the 7-DoF delta end-effector pose of the arm (including x, y, z, roll, pitch, yaw and gripper opening). The entire policy network is trained end-to-end using the imitation learning objective and we specify the details of the objective in the Appendix (Equation 1). Importantly, the VLM used to detect the objects is frozen during training, so that it does not overfit to the objects in the robot demonstration data. The policy is trained with this frozen VLM in the loop, so that the policy can learn to be robust to errors made by the VLM given other information in the image.

### 3.4 Practical Implementation

To detect objects in our robot images, we use the Owl-ViT open-vocabulary object detector [43]. In practice, we find that it is capable of detecting most clearly visible objects without any fine-tuning, given a descriptive natural language phrase. The interface to the detector requires a natural language phrase describing what to detect (e.g., "An image of a small blue elephant toy.") along with an image to run the detection on. The output from the model is a score map indicating which locations are most likely to correspond to the natural language description and their associated bounding boxes. We select a universal score threshold to filter detections. To detect our objects, we rely on some prompt-engineering using descriptive phrases including the color, size, and shape of objects, though most of our prompts worked well on the first attempt. We share the prompts in the Appendix.

To make the inference time practical on real robots, we extract the object information via VLM only in the first episode frame. By doing so, most of the heavy computation is executed only once at the beginning and we can perform real-time control for the entire episode. Since the information is appended to the current observation, we rely on the policy to find the corresponding object in the current image if the object has moved since the first timestep.

### 3.5 Training Data

We start with the demonstration data used by RT-1 [24] covering 16 unique objects. Despite the use of the VLM for semantic generalization, we expect that the policy will require more physical object diversity to generalize to novel objects. Therefore, we expand the dataset with additional diverse "pick" data across a set of 90 diverse objects, for a total of 106 objects, as shown in Figure 3. We choose to only expand the set of objects for the picking skill, since it is the fastest skill to execute and therefore allows for the greatest amount of diverse data collection within a limited budget of demonstrator time. Our additional set of 90 diverse objects only appear in "pick" episodes. All other tasks, such as "move near" or "place into", must be learned from the original 16 objects in the RT-1 dataset. Detailed statistics are in Appendix Figure 9.

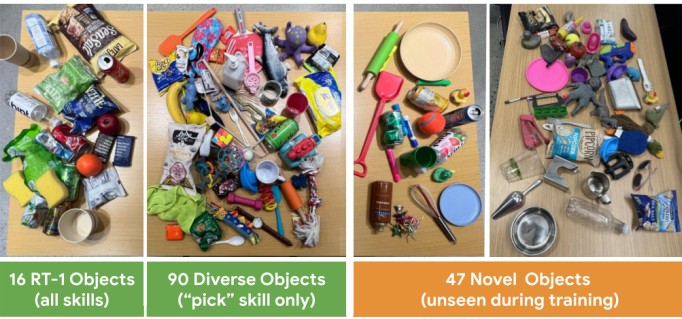

Figure 3: (Left) RT-1 objects that account for $\approx 70\%$ of training data covering all skills. (Middle) Diverse training objects that appear only in "pick" demonstrations. (Right) Unseen objects used only for evaluation.

## 4 Experiments

Our experiments answer the following questions: 1) Does MOO generalize across objects for different skills including unseen objects? 2) Does MOO generalize beyond new objects – Is MOO robust to distractors, backgrounds and environments? 3) Can the intermediate representation used support non-linguistic modalities to specify the task? 4) Does the object generalization performance scale with the (a) number of training episodes, (b) number of unique objects in the training episodes and (c) size of the model? 5) Can MOO be used for open-world navigation and manipulation?

### 4.1 Experimental Setup

**Seen and unseen objects.** The training data is collected with teleoperation on table-top environments across a set of 106 different object types. We evaluate performance on a subset with 49 objects "seen" in training and report the performance as "seen". We hold out 47 objects not present in training and report performance on these as "unseen". These 47 held-out objects are comprised of 22 objects of the categories seen in training and 25 objects of unseen categories. These objects are listed in Appendix Table 1. Note that previous works often focus on unseen combinations of previously seen commands and objects (e.g. "pick an apple" even though the training data contains "move an apple into a bowl" and "pick a bowl"); we adopt a more strict definition of unseen objects, where our unseen objects were not seen in the robot's training demonstration data at any point for any task, therefore making our unseen performance a zero-shot object generalization problem. Furthermore, we report results across different environments that introduce novel textures, backgrounds, locations, and additional open-world objects not present in the training data.

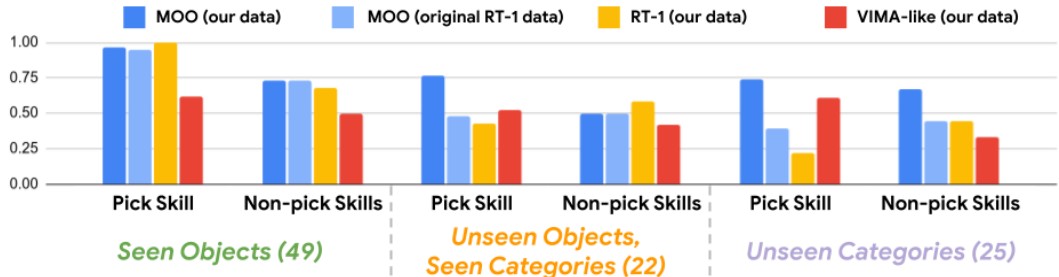

Figure 4: **Main Results.** While baseline methods perform competitively on in-distribution combinations of objects and skills seen during training, they fail to generalize to novel objects. MOO substantially improves generalization to novel objects, especially those in unseen categories and for the "pick" skill.

**Evaluation details.** We evaluate on a set of tabletop tasks involving manipulating a diverse set of objects. We use mobile manipulators with 7 degree-of-freedom arm and two-fingered gripper (Appendix Figure 10). Our experiments evaluate the percent of successfully completed manipulation commands which include five skills: "pick", "move near", "knock," "place upright," and "place into" across a set of evaluation episodes (definition and success criteria follow RT-1 [24] and are described in the Appendix). To study object specificity and robustness, for all evaluation episodes, we include between two to four distractor objects in the scene which the robot should not interact with. For each evaluation episode, we randomly scatter the evaluation object(s) and the distractor objects onto the table. There is no consistent placement of the target object relative to the distractors. We repeat this process 21 times and report the performance. We present the experimental setup in Appendix Figure 10.

**Baselines.** We compare two prior methods: RT-1 [24] and a modified version of VIMA [25], referred to as "VIMA-like". VIMA-like preserves the cross-attention mechanism, but uses the mask image as the prompt token and the current image as state token. These modifications are necessary because VIMA uses Transporter-based action space and is not applicable to our task, i.e., our robot arm moves in 6D and has a gripper that can open and close continuously. These two baselines correspond to common alternatives where the computer vision data is used as a pre-training mechanism (as in RT-1) or object-centric information is fed to the network through cross attention (as in VIMA-like).

## 4.2 Experimental Results

**Generalization to Novel Objects.** We investigate the question: *Does MOO generalize across objects for different manipulation skills including objects never seen at training time?* Experiments are presented in Figure 4 and example trajectories are shown in Appendix Figure 12. Relative to the baselines on the pick tasks, MOO exhibits substantial improvement over the seen object performance as well as the unseen objects, which in both cases reaches $\sim 50\%$ improvement. MOO can correctly utilize a VLM to find novel objects and incorporate that information more effectively than the VIMA-like

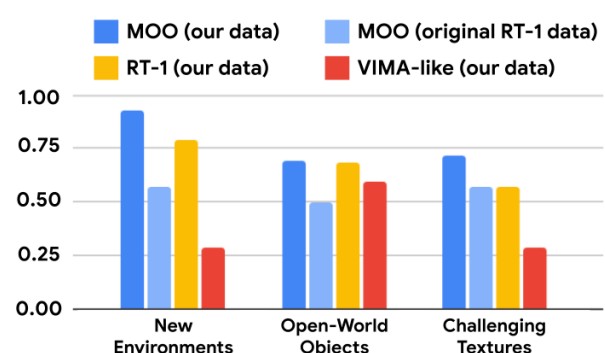

Figure 5: MOO is able to generalize to new objects, textures, and environments with greater success than prior methods. Visualizations are shown in Figure 6.

baseline. When comparing the performance on seen objects for the skills other than *pick*, we observe a slightly worse performance than for the *pick* tasks. This is understandable since the "Seen objects" for "Non-pick skills" have only been seen during the *pick* episodes as shown in Appendix Figure 9. This demonstrates MOO's ability to transfer the learned object generalization across the skills so that the objects that have only been picked can now be also used for other skills. In addition, MOO exhibit generalization to unseen objects (i.e. unseen in any previous tasks, including pick) that is at the same level as for unseen objects for the pick skill, and $50\%$ better than baseline.

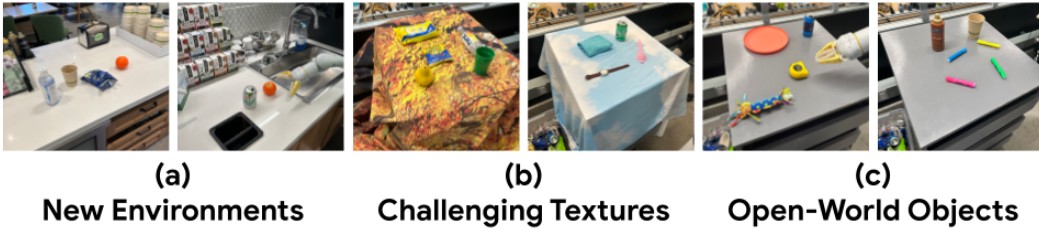

**(a) New Environments**  **(b) Challenging Textures**  **(c) Open-World Objects**

Figure 6: To study the robustness of MOO, we evaluate on (a) new environments, (b) challenging texture backgrounds which are visually similar to unseen objects in the scene, and (c) additional open-world objects.

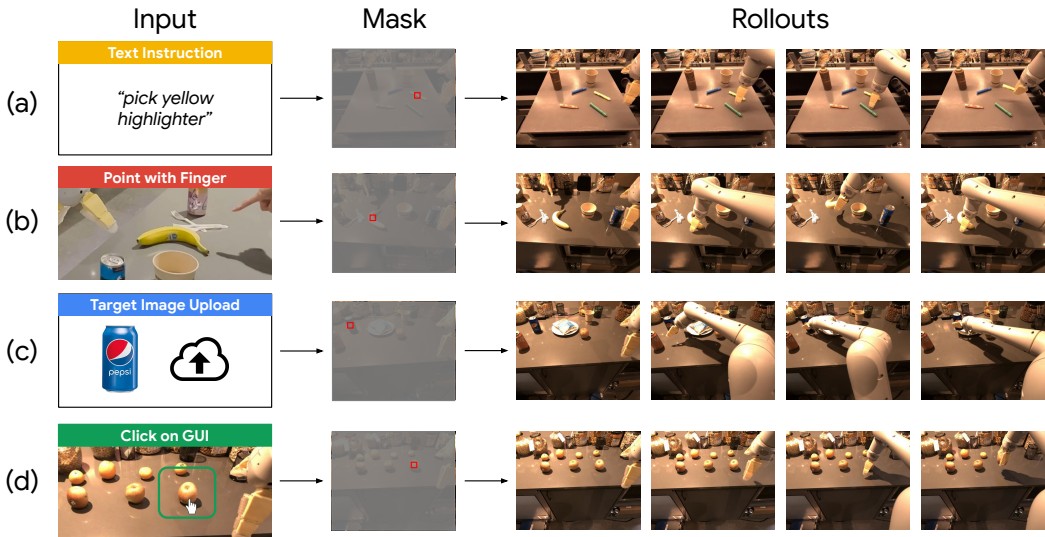

Figure 7: We explore using various input modalities to generate the single-pixel object representations used by MOO. (a) shows the standard mask generation process using OWL-ViT with a text instruction. (b) shows using a VLM to generate a text caption, then fed to OWL-ViT. (c) shows an uploaded image to prompt OWL-ViT. (d) shows a user providing a ground-truth mask via a GUI.

**Robustness Beyond New Objects.** To further test the robustness of MOO, we analyze novel evaluation settings with significantly increased difficulty and visual variation, which are shown in Figure 6. To reduce the number of real robot evaluations, we focus this comparison on the picking skill. The results are presented in Figure 5. Across these challenging evaluation scenes, MOO is significantly more robust compared to VIMA-like [25] and RT-1 [24]. This indicates that the use of VLMs in MOO not only improves generalization to new objects that the robot has not interacted with, but also significantly improves generalization to new backgrounds and environments.

**Input Modality Experiments.** To answer our third question, we perform a number of qualitative experiments testing different input modalities (detailed description in the Appendix). We find that MOO is able to generalize to masks generated from a variety of upstream input modalities, even under scenarios outside the training distribution including scenes with duplicate objects and clutter.

As the first qualitative example, Figure 7(b) illustrates that VLM such as PaLI [51] can infer what object a human is pointing at, allowing OWL-ViT to generate an accurate mask of the object of interest. Secondly, OWL-ViT can also use visual query features instead of textual features to generate a mask, enabling images of target objects to act as conditioning for MOO, as shown in Figure 7(c). This modality is useful in cases where text-based mask generation due to ambiguity in natural language, or when target images are found in other scene contexts. We explore both the setting where target images are sourced from similar scenes or from diverse internet images. Finally, we show that MOO can interpret masks directly provided by humans via a GUI, as shown in Figure 7(d). This is useful in cases where both text-based and image-based mask generation is difficult, such as with duplicate or cluttered objects. MOO is robust to how upstream input masks were generated, and our preliminary results suggest interesting future avenues in the space of human-robot interaction.

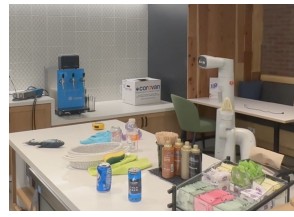 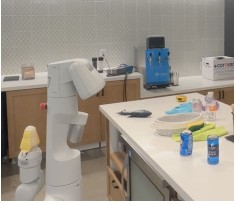 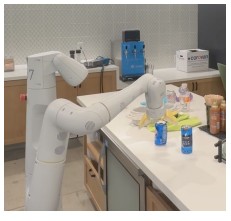 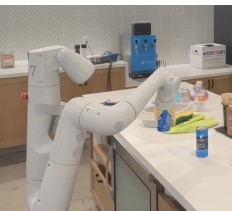

CoW: *"find the pepsi"* | ├──────────── MOO: *"pick up the pepsi"* ────────────┤

Figure 8: We present CoW-MOO, a system that combines an open-vocabulary object navigation by CoW [52] with open-world manipulation by MOO. Full videos are shown on the project's website.

**MOO Ablations.** We conduct a number of ablations to assess the impact of the size and diversity of our dataset and the scale (in terms of number of parameters) of our model. In Appendix Table 3 we vary both the number of unique objects in the training set (reducing it from 106 to 53 to 16 unique training objects) and the number of total training episodes (reducing it by half – from 59051 training episodes to 29525) while keeping all objects in the dataset. We aim to vary these two axes independently to determine the impact of the overall size of the dataset vs its object diversity on the final results. Interestingly, we find that seen object performance is not affected by reducing object diversity, but generalization to unseen objects is very sensitive to object diversity.

Additionally, we investigate the impact of scaling model size. We train two smaller versions of MOO where we scale down the total number of layers and the layer width by a constant factor. The version of MOO that we use in our main experiments has 111M parameters, which, for the purpose of this ablation, we then reduce by an order of magnitude down to 10.2M and then by 5X again down to 2.37M. Comparing different sizes of the model, we find significant drop offs in both "seen" (from 98% to 54% and 39% respectively) and "unseen" object performance (from 79% to 50% and 13%; see Appendix Figure 11 for a graph of the results). We also note that we could not make MOO larger than 111M parameters without increasing the latency on robot to an unacceptable level, but we expect continued performance gains with bigger models if latency requirements can be relaxed.

**Open-World Navigation and Manipulation.** Finally, we consider how such a system can be integrated with open-vocabulary object-based navigation. Coincidentally, there is an open-vocabulary object navigation algorithm called Clip on Wheels (CoW) [52]; we implement a variant of CoW and combine it with MOO, which we refer to as CoW-MOO. CoW handles open-vocabulary navigation to an object of interest, upon which MOO continues with manipulating the target object. This combination enables a truly open-world task execution, where the robot is able to first find an object it has never interacted with, and then successfully manipulate it to accomplish the task. We show example qualitative experiments in Figure 8 and in the video of this system on the project's website[2].

## 5 Conclusion and Limitations

In this paper we presented MOO, an approach for leveraging the rich semantic knowledge captured by vision-language models in robotic manipulation policies. We conduct $1,472$ real world evaluations to show that MOO allows robots to generalize to novel instructions involving novel objects, enables greater robustness to visually challenging table textures and new environments, is amenable to multiple input modalities, and can be combined with open-vocabulary semantic navigation.

Despite the promising results, MOO has multiple important limitations. First, the object mask representation used by MOO may struggle in visually ambiguous cases, such as where objects are overlapping or occluded. Second, we expect the generalization of the policy to still be limited by the motion diversity of training data. For example, we expect that the robot may struggle to grasp novel objects with drastically different shapes or sizes than those seen in the training demonstration data, even with successful object localization. Third, instructions are currently expected to conform to a set of templates from which target objects and verbs can be easily separated. We expect this limitation could be lifted by leveraging an LLM to extract relevant properties from freeform instructions. Finally, MOO cannot currently handle complex object descriptions involving spatial relations, such as "the small object to the left of the plate." Fortunately, we expect performance on tasks such as these to improve significantly as vision-language models continue to advance moving forward.

---

[2]https://robot-moo.github.io/

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

## Appendix

**Imitation Learning and RT-1**

MOO builds upon a language-conditioned imitation learning setup. The goal of language-conditioned imitation learning is to learn a policy $\pi(a \mid \ell, o)$, where $a$ is a robot action that should be applied given the current observation $o$ and task instruction $\ell$. To learn a language-conditioned policy $\pi$, we build on top of RT-1 [24], a recent robotics transformer-based model that achieves high levels of performance across a wide variety of manipulation tasks. RT-1 uses behavioral cloning [53], which optimizes $\pi$ by minimizing the negative log-likelihood of an action $a$ given the image observations seen so far in the trajectory and the language instruction, using a demonstration dataset containing $N$ demonstrations:

$$J(\pi) := \sum_{n=1}^{N} \sum_{t=1}^{T^{(n)}} \log \pi(a_t^{(n)} \mid \ell^{(n)}, \{o_j^{(n)}\}_{j=1}^{t}). \tag{1}$$

**Vision-Language Models**

In recent years, there has been a growing interest in developing models that can detect objects in images based on natural language queries. These models, known as vision-language models (VLMs), are enabling detectors to identify a wide range of objects based on natural language queries. Typically the text queries are tokenized and embedded in a high-dimensional space by a pre-trained language encoder, and the image is processed by a separate network to extract image features into the same embedding space as the text features. The language and image representations are then combined to make predictions of the bounding boxes and segmentation masks. Given a natural language query, $q$, and an image observation on which to run detection, $o$, these models aim to produce a set of embeddings for the image $f_i(o)$ with shape $(\text{height}, \text{width}, \text{feature dim})$ and an embedding of the language query $f_l(q)$ with shape feature dim such that $\text{logits} = f_i(o) \cdot f_l(q)$ gives a logit score map and is maximized at regions in $o$ which correspond to the queries in $q$. Each image embedding location within $f_i(o)$ is also associated with a predicted bounding box or mask indicating the spatial extent of that object corresponding to $f_i(o)$. In this work, we use the Owl-ViT detector [43], which we discuss further in Sec. 3.4.

**Datasets**

We collect a teleoperated demonstration data that focuses on increasing object diversity for the most efficient skill to collect data for, the picking task. This dataset of 13,239 episodes was collected with a similar procedure to [24], with expert users utilizing Oculus Virtual Reality controllers for teleoperation. Detailed dataset statistics are in Figure 9 and Table 1.

| Object | Included in Training | Included in Evaluation | | |
|---|---|---|---|---|
| | | Seen Object | Unseen Object, Seen Category | Unseen Category |
| red grapefruit can | yes | | | |
| coke zero can | yes | | | |
| pineapple spindrift can | yes | | | |
| lemon spindrift can | yes | | | |
| love kombucha | yes | | | |
| original pepper can | yes | | | |
| fruit gummies | yes | | | |
| instant oatmeal pack | yes | | | |
| brie cheese cup | yes | | | |
| coffee mixing stick | yes | | | |
| white sparkling can | yes | | | |
| diet pepper can | yes | | | |
| lemon sparkling water can | yes | | | |
| black pen | yes | | | |
| orange plastic bottle | yes | | | |
| blue pen | yes | | | |
| coffee cup sleeve | yes | | | |
| regular 7up can | yes | | | |
| small salmon plate | yes | | | |
| diet coke can | yes | | | |
| lemonade plastic bottle | yes | | | |
| original redbull can | yes | | | |
| numi tea bag | yes | | | |
| popcorn chip bag | yes | | | |
| cereal scoop | yes | | | |
| blackberry hint water | yes | | | |
| green cookies bag | yes | | | |
| watermelon hint water | yes | | | |
| spoon | yes | | | |
| coffee cup lid | yes | | | |
| green pear | yes | | | |
| coffee cup | yes | | | |
| iced tea can | yes | | | |
| ito en green tea | yes | | | |
| pink lunch box | yes | | | |
| chocolate caramel candy | yes | | | |
| small beige plate | yes | | | |
| large yellow spatula | yes | | | |
| large hot pink plate | yes | | | |
| red bowl | yes | | | |
| green bowl | yes | | | |
| orange spatula | yes | | | |
| large blue plate | yes | | | |
| large baby pink plate | yes | | | |
| small purple plate | yes | | | |
| small blue spatula | yes | | | |
| small green plate | yes | | | |
| table tennis paddle | yes | | | |
| green brush | yes | | | |
| rubiks cube | yes | | | |
| gray suction toy | yes | | | |
| toy ball with holes | yes | | | |
| large tennis ball | yes | | | |
| gray microfiber cloth | yes | | | |
| toy boat train | yes | | | |

| | | |
|---|---|---|
| teal and pink toy car | yes | |
| dna chew toy | yes | |
| slinky toy | yes | |
| raspberry baby teether | yes | |
| small purple spatula | yes | |
| milano dark chocolate | yes | |
| badminton shuttlecock | yes | |
| chain link toy | yes | |
| orange cup | yes | |
| head massager | yes | |
| square cheese | yes | |
| boiled egg | yes | |
| blue cup | yes | |
| chew toy | yes | yes |
| fish toy | yes | yes |
| egg separator | yes | yes |
| blue microfiber cloth | yes | yes |
| yellow pear | yes | yes |
| small orange rolling pin | yes | yes |
| wrist watch | yes | yes |
| pretzel chip bag | yes | yes |
| disinfectant wipes | yes | yes |
| pickle snack | yes | yes |
| octopus toy | yes | yes |
| catnip toy | yes | yes |
| orange | yes | yes |
| 7up can | yes | yes |
| apple | yes | yes |
| coke can | yes | yes |
| swedish fish bag | yes | yes |
| large green rolling pin | yes | yes |
| place green can upright | yes | yes |
| black sunglasses | yes | yes |
| blue chip bag | yes | yes |
| pepsi can | yes | yes |
| pink shoe | yes | yes |
| blue plastic bottle | yes | yes |
| green can | yes | yes |
| orange can | yes | yes |
| water bottle | yes | yes |
| redbull can | yes | yes |
| green jalapeno chip bag | yes | yes |
| rxbar chocolate | yes | yes |
| rxbar blueberry | yes | yes |
| brown chip bag | yes | yes |
| green rice chip bag | yes | yes |
| sponge | yes | yes |
| chocolate peanut candy | yes | yes |
| banana | yes | yes |
| oreo | yes | yes |
| cheese stick | yes | yes |
| yellow bowl | yes | yes |
| large green plate | yes | yes |
| white coat hanger | yes | yes |
| green microfiber cloth | yes | yes |
| small blending bottle | yes | yes |
| floral shoe | yes | yes |
| dog rope toy | yes | yes |
| red cup | yes | yes |

| | | | | |
|---|---|---|---|---|
| fork | yes | yes | | |
| disinfectant pump | yes | yes | | |
| blue balloon | yes | yes | | |
| bird ornament | | | | yes |
| red plastic shovel | | | | yes |
| whisk | | | | yes |
| baby toy | | | | yes |
| brown dinosaur toy | | | | yes |
| pikmi pops confetti toy | | | | yes |
| white marker holder | | | | yes |
| white toilet scrub | | | | yes |
| pink stapler | | | | yes |
| green dolphin toy | | | | yes |
| purple eggplant | | | | yes |
| small green dinosaur toy | | | | yes |
| green blocks | | | | yes |
| navy toy gun | | | | yes |
| gray pouch | | | | yes |
| small red motorcycle toy | | | | yes |
| bike pedal | | | | yes |
| c clamp | | | | yes |
| burgundy paint brush | | | | yes |
| transparent hint water bottle | | | | yes |
| shiny steel coffee grinder holder | | | | yes |
| transparent plastic cup | | | | yes |
| shiny steel mug | | | | yes |
| shiny steel scooper | | | | yes |
| shiny pink steel bowl | | | | yes |
| light pink sunglasses | | | yes | |
| pink marker | | | yes | |
| cold brew can | | | yes | |
| ginger lemon kombucha | | | yes | |
| green cup | | | yes | |
| green sprite can | | | yes | |
| large orange plate | | | yes | |
| pineapple hint water | | | yes | |
| small blue plate | | | yes | |
| small hot pink plate | | | yes | |
| black small duck | | | yes | |
| small purple bowl | | | yes | |
| purple toy boat | | | yes | |
| black chip bag | | | yes | |
| teal sea salt chip bag | | | yes | |
| blue sea salt chip bag | | | yes | |
| sea salt seaweed snack | | | yes | |
| gray sponge | | | yes | |
| red velvet snack bar | | | yes | |
| red bell pepper | | | yes | |
| blue toy boat train | | | yes | |
| green tennis ball | | | yes | |

Table 1: List of objects used in training and evaluation. There are 3 types of objects used in evaluation: 49 objects which were seen in training, 22 unseen objects of categories which were seen in training, 25 unseen objects of unseen categories.

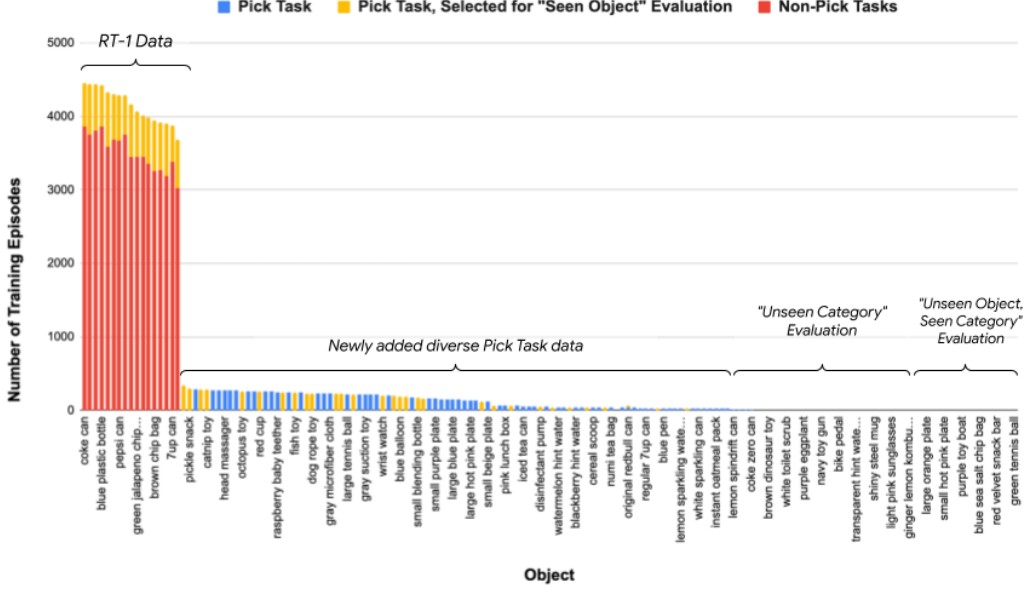

Figure 9: Distribution of training objects across the training dataset. We augmented RT-1 data (on the left) with a large number of diverse pick episodes (in the middle) in order to demonstrate strong generalization to unseen objects (on the right). Blue and yellow bars represent "pick" episodes and red bars represent other tasks like "move near" or "knock." Yellow bars portray the objects randomly selected for "Seen Object" evaluations. Objects for "Unseen Category" and "Unseen Object, Seen Category" evaluations are shown to the right.

## Experiments

We show a visualization of our 7-DoF manipulation robot in Figure 10.

**Skills.** Our experiments evaluate the percent of successfully completed manipulation commands which include five skills: "pick", "move near", "knock," "place upright," and "place into" across a set of evaluation episodes. The definition of the tasks follows RT-1 [24]: For "pick", success is defined as (1) grasping the specified object and (2) lifting the object at least 6 inches from the table top. For "move near", success is defined as (1) grasping the specified object and (2) placing it within 6 inches of the specified target object. For "knock", success is defined as placing the specified object from an "upright" position onto its side. "Place upright" tasks are the inverse of "knock" and involve placing an object from its side into an upright position. Finally, "place into" tasks involve placing one object into another, such as an apple into a bowl.

**Robustness evaluation details.** We evaluate the robustness of MOO on a variety of visually challenging scenarios with drastically different furniture and backgrounds, as shown in Figure 6; the results are reported in Figure 5. The first set of these difficult evaluation scenes introduces six evaluations across five additional open-world objects that correspond to various household items that have not been seen at any point during training. The second set of difficult scenes introduces 14 evaluations across two patterned tablecloths; these tablecloth textures are significantly more challenging than the plain gray counter-tops seen in the training demonstration dataset. Finally, the last set of difficult scenes include 14 evaluations across three new environments in natural kitchen and office spaces that were never present training. These new scenes simultaneously change the counter-top materials, backgrounds, lighting conditions, and distractor items.

**Input modality demonstration details.** We explore the ability of MOO to incorporate object-centric mask representations that are generated via different processes than the one used during training. During training, an OWL-ViT generates mask visual representations from textual prompts, as described in Section 3.2. We study whether MOO can successfully accomplish manipulation tasks given (1) a mask generated from a text caption from a generative VLM, (2) a mask generated from an image query instead of a text query, or (3) a mask directly provided by a human via a GUI. For each of these cases, we implement different procedures for generating the object mask representation, which are then fed to the frozen MOO policy.

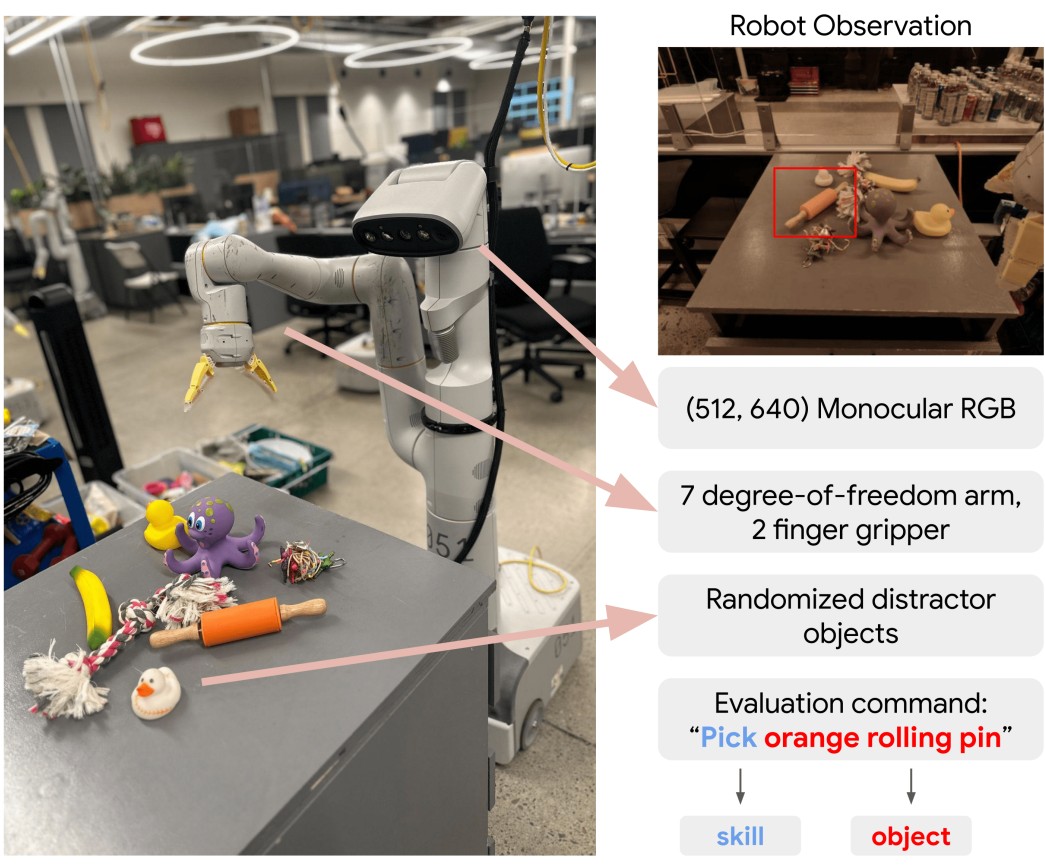

Figure 10: Image of our robot hardware and evaluation setting.

**Mask Sensitivity Study** We originally noted the qualitative observation that MOO seemed to exhibit robustness to imperfect object-centric mask localization. We ran an additional experiment of evaluating a MOO policy with masks that contain artificially added localization noise. We study four cases where we ablate the mask while keeping the starting scene the same: (1) the baseline when the mask is at the centroid of the object, (2) when the mask is still on the object of interest but not at the centroid, (3) when the mask is off of the object entirely but still within roughly 5cm, and (4) when the mask is far and more than 5cm from the object of interest. We add the artificial noise manually by starting with the centroid mask and then manually ablating the masks. We run a total of 20 trials across 5 tasks involving one seen object (green rolling pin), one unseen object in a seen category (cold brew can), and three unseen object categories (egg plant, shiny sunglasses, transparent bottle). We find that performance degrades as more noise is added: case (1) achieves 5/5 successes, case (2) achieves 4/5 successes, case (3) achieves 3/5 successes, and case (4) achieves 3/5 successes. Qualitatively, we observe that the policy sometimes initially reaches for the inaccurate mask location, and is sometimes able to recover and re-approach the correct target object. In one notable example, the policy faced an off-center mask on the lower part of the clear bottle (a visually challenging object, since no transparent objects were in the training set) and grasped the clear bottle in the upper section, which is the correct strategy for re-orienting the bottle upright. Additionally, a few examples showed that the policy was able to retry after failures caused by inaccurate masks; this suggest that the policy does not just memorize going to the location of the mask, but instead also pays attention to semantics. We provide the table of quantitative results in Table 2.

**Training data ablation.** We ablate the amount of data used to train MOO, and find that both data diversity and data scale are important, as shown in Table 3.

**Prompts used**

We use the following prompts to OWL-ViT detect our objects. All prompts were prefixed with the phrase "An image of a".

| Mask Ablation Amount | Pick Skill | | | Upright Skill | Knock Skill |
|---|---|---|---|---|---|
| | Green Rolling Pin | Eggplant | Shiny Glasses | Clear Bottle | Cold Brew Can |
| baseline centroid mask | 1 | 1 | 1 | 1 | 1 |
| off-center on-object mask | 0 | 1 | 1 | 1 | 1 |
| off-object less than 5cm mask | 1 | 1 | 1 | 0 | 0 |
| off-object more than 5cm mask | 1 | 0 | 1 | 0 | 1 |

Table 2: We evaluate a MOO policy for 20 trials with masks that contain artificially added localization noise. Notably, we find that MOO is able to recover from misspecified masks which may not be centered on the target object, for both seen objects and novel objects altogether.

| Dataset Filtering | | Pick | |
|---|---|---|---|
| Objects | Episodes per Object | Seen objects | Unseen objects |
| 100% | 100% | **98** | **79** |
| 50% | 100% | 92 | 75 |
| 18% | 100% | 88 | 19 |
| 100% | 50% | 46 | 38 |
| 100% | 10% | 23 | 0 |

Table 3: Performance of MOO in percentage of success relative to the amount of data used for training. Both data scale and data diversity are important.

7up can → "white can of soda"
banana → "banana"
black pen → "black pen"
blue chip bag → "blue bag of chips"
blue pen → "blue pen"
brown chip bag → "brown bag of chips"
cereal scoop → "cereal scoop"
chocolate peanut candy → "bag of candy snack"
coffee cup → "coffee cup"
coke can → "red can of soda"
coke zero can → "can of soda"
disinfectant pump → "bottle"
fork → "fork"
green can → "green aluminum can"
green cookies bag → "green snack food bag"
green jalapeno chip bag → "green bag of chips"
green sprite can → "green soda can"
knife → "knife"
orange can → "orange aluminum can"
orange plastic bottle → "orange bottle"
oreo → "cookie snack food bag"
pepsi can → "blue soda can"
popcorn chip bag → "bag of chips"
pretzel chip bag → "bag of chips"
red grapefruit can → "red aluminum can"
redbull can → "skinny silver can of soda"
rxbar blueberry → "small blue rectangular snack food bar"
spoon → "spoon"
swedish fish bag → "bag of candy snack food"
water bottle → "clear plastic waterbottle with white cap"
white sparkling can → "aluminum can"
blue plastic bottle → "clear plastic waterbottle with white cap"
diet pepper can → "can of soda"
disinfectant wipes → "yellow and blue pack"
green rice chip bag → "green bag of chips"
orange → "round orange fruit"
paper bowl → "round bowl"
rxbar chocolate → "small black rectangular snack food bar"
sponge → "scrub sponge"
blackberry hint water → "clear plastic bottle with white cap"
pineapple hint water → "clear plastic bottle with white cap"

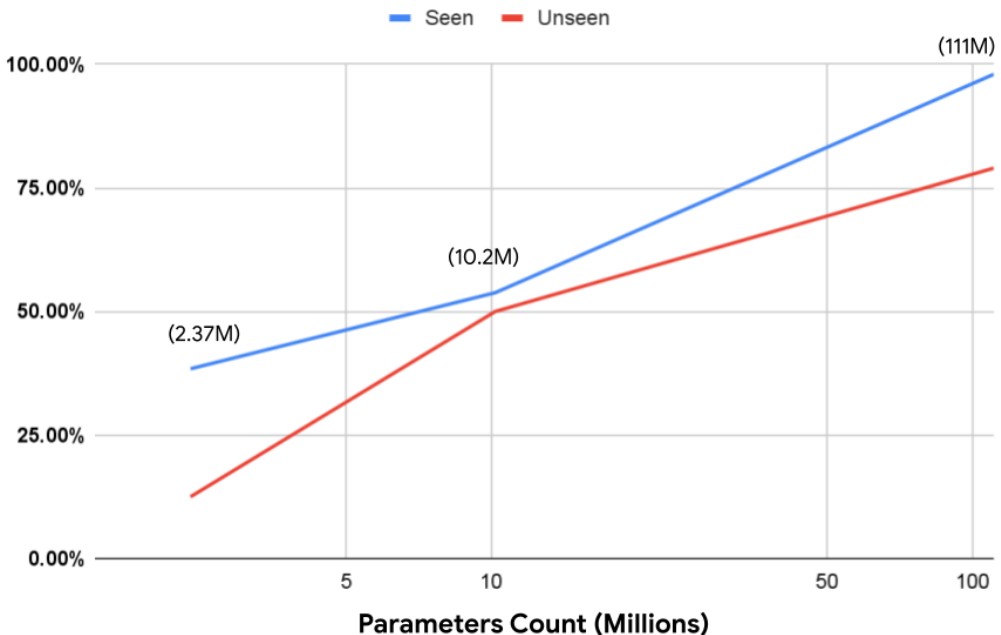

Figure 11: Pick success vs. model size. We see continuous improvements on both seen and unseen objects as we increase the number of parameters of our model architecture while keeping the data set size fixed. In comparison to our main model, we scaled down layer widths and depth by the same constant multiplier. We expect more performance gains at larger model capacity, yet are currently unable to scale further due to real time inference constraints on our robot.

watermelon hint water → "clear plastic bottle with white cap"
regular 7up can → "can of soda"
lemonade plastic bottle → "clear plastic bottle with white cap"
diet coke can → "silver can of soda"
yellow pear → "yellow pear"
green pear → "green pear"
instant oatmeal pack → "flat brown pack of instant oatmeal"
coffee mixing stick → "small thin flat wooden popsicle stick"
coffee cup lid → "round disposable coffee cup lid"
coffee cup sleeve → "brown disposable coffee cup sleeve"
numi tea bag → "small flat packet of tea"
fruit gummies → "small blue bag of snacks"
chocolate caramel candy → "small navy bag of candy"
original redbull can → "can of energy drink with dark blue label"
cold brew can → "blue and black can"
ginger lemon kombucha → "yellow and tan aluminum can with brown writing"
large orange plate → "circular orange plate"
small blue plate → "circular blue plate"
love kombucha → "white and orange can of soda"
original pepper can → "dark red can of soda"
ito en green tea → "light green can of soda"
iced tea can → "black can of soda"
cheese stick → "yellow cheese stick in wrapper"
brie cheese cup → "small white cheese cup with wrapper"
pineapple spindrift can → "white and cyan can of soda"
lemon spindrift can → "white and brown can of soda"
lemon sparkling water can → "yellow can of soda"
milano dark chocolate → "white pack of snacks"
square cheese → "small orange rectangle packet "
boiled egg → "small white egg in a plastic wrapper"
pickle snack → "small black and green snack bag"

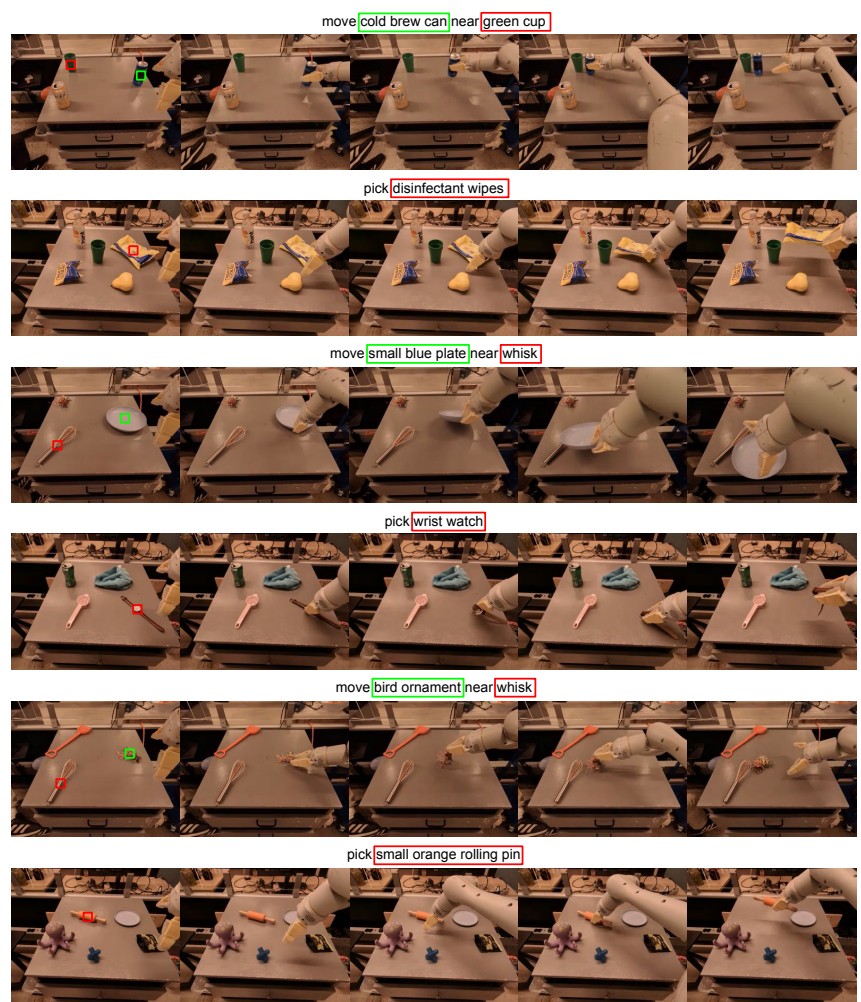

Figure 12: Example images of our policy detecting and grasping objects not seen during training time. The object detections are colored in correspondence to the text above the image, and the images are ordered left to right across time.

red cup → "plastic red cup"
blue cup → "plastic blue cup"
orange cup → "plastic orange cup"
green cup → "plastic green cup"
head massager → "metal head massager with many wires"
chew toy → "blue and yellow toy with orange polka dots"
wrist watch → "wrist watch"
small orange rolling pin → "small orange rolling pin with wooden handles"
large green rolling pin → "large green rolling pin with wooden handles"
rubiks cube → "rubiks cube"
blue microfiber cloth → "blue cloth"
gray microfiber cloth → "gray cloth"
green microfiber cloth → "green cloth"
small blending bottle → "small turqoise and brown bottle"
large tennis ball → "large tennis ball"
table tennis paddle → "table tennis paddle"
octopus toy → "purple toy octopus"
pink shoe → "pink shoe"
floral shoe → "red and blue shoe"
whisk → "whisk"
orange spatula → "orange spatula"

small blue spatula → "small blue spatula"
large yellow spatula → "large yellow spatula"
egg separator → "large pink cooking spoon"
green brush → "green brush"
small purple spatula → "small purple spatula"
badminton shuttlecock → "shuttlecock"
black sunglasses → "black sunglasses"
toy ball with holes → "toy ball with holes"
red plastic shovel → "red plastic shovel"
bird ornament → "colorful ornament with blue and yellow confetti"
blue balloon → "blue balloon animal"
catnip toy → "small dark blue plastic cross toy"
raspberry baby teether → "red and green baby pacifier"
slinky toy → "gray metallic cylinder slinky"
dna chew toy → "big orange spring"
gray suction toy → "gray suction toy"
teal and pink toy car → "teal and pink toy car"
two pound purple dumbbell → "purple dumbbell"
one pound pink dumbbell → "pink dumbbell"
three pound brown dumbbell → "brown dumbbell"
dog rope toy → "white pink and gray rope with knot"
fish toy → "fish"
chain link toy → "skinny green rectangular toy"
toy boat train → "plastic toy boat"
white coat hanger → "white coat hanger"

