# OpenReview forum: "Open-World Object Manipulation using Pre-Trained Vision-Language Models"
_robot-learning.org/CoRL/2023/Conference — CoRL 2023 Poster_

### Official Review · Reviewer_VQYg · 2023-07-14

**Confidence:** 5
**Originality:** Fair
**Technical Quality:** Good
**Clarity Of Presentation:** Fair
**Impact:** 3

**Recommendation:**

Weak Accept: I recommend accepting the paper, but will not argue for my recommendation if the majority of other reviewers have a different opinion.

**Review:**

The proposed extension to the RT-1 architecture appears to be simple yet effective. Moreover, specifying the target skill with just the verb while indicating the target object on which the skill should be executed is quite appealing and intuitive and is the right step in the direction of developing/learning general skills that can operate on unseen object categories.

My main concern is that the proposed architecture does not depend on depth information. Although the authors see this as an advantage compared to prior work, depth is fundamental to manipulating previously unseen objects without prior knowledge. To see why this is the case, consider attempting to execute a task "pick cube" where the cube is a previously unseen object. Furthermore, consider a setting where the gripper is directly above the cube and all the robot has to do is move the gripper downwards and close it to pick up the cube. Given an RGB image of the gripper above the cube, without prior knowledge, there is no way to infer whether the gripper is vertically aligned with the cube, and whether the policy should move the gripper forwards, backwards or downwards. This is because either:

- The gripper is already aligned.
- The cube is quite small, and the gripper has to be moved closer to the robot to align vertically with the cube.
- The cube is quite large, and the gripper has to move further away from the robot to align vertically with the cube.

Of course, some of this ambiguity is partially solved with the camera's perspective. However, depth is fundamental to manipulating previously unseen objects in general; otherwise, this becomes an ill-defined problem.

Now, in the experiments, it is shown that the proposed policy can successfully pick many previously unseen objects. However, I believe this is because the objects considered are similar to those used during training. Although the authors explicitly wrote, "...we adopt a more strict definition of unseen objects, where our unseen object categories were not seen in the robot's training demonstration data at any point for any task", closely examining Figure 3 reveals that many of the object categories present in the novel objects can also be seen in the training objects. For instance, there is a green rolling pin in the novel objects and an orange rolling pin in the training objects. Cans and bottles can be seen in the novel and training objects. The same holds for bath ducks, bottles and food packaged in foil bags (e.g. bags of crisps). Hence, it is likely that the policy can only manipulate the considered "novel" objects without depth information because it has seen training data for similar objects with similar dimensions.

**Quality Of The Limitations Section:**

Limitations are addressed clearly

**Questions For Rebuttal:**

- In Section 3, you highlight that "The key design choice of MOO lies in how to represent object information encoded in VLMs and how to feed that information to the instruction-conditioned policy." Have you maybe tried representing this information as a segmentation mask, for example, by combining an open vocabulary object detection model with a segmentation model, or as a bounding box? And do you have any insights on how this affects the performance? Given that this representation is a key design choice, I wonder if the choice of a one-hot pixel representation was derived from experiments or if it was the only representation attempted.
- In Section 4.1, you wrote, "We hold out 47 objects not present in training and report performance on these as "unseen"". However, Figure 9 shows only 13 objects "Selected for "unseen" evaluation". Moreover, in the evaluation details, it is written that the experimental process is repeated 21 times to report the performance. Can you clarify how many unseen objects the methods were tested on and which are these objects?
- The evaluation details could be clearer as it is unclear how many evaluations were conducted for each of the experiments, whether every unseen object was considered from the ones shown in Figure 3, or whether they were sampled from all the objects independently for each method.

**Robotics Focus:**

Sufficient demonstration on hardware

**Summary Of Paper:**

The authors propose a simple modification to the RT-1 architecture, that extends the input to also take a mask indicating a pixel(s) that roughly lies at the centre of a target object(s) and change the input language instruction from a full sentence specifying what task to complete (e.g. "pick banana") to a verb associated with the target skill (e.g. "pick"). The authors also extend the training dataset of RT-1 to contain additional demonstrations of the "pick" skill for 90 novel objects and train the modified RT-1 architecture on the original RT-1 data plus their newly collected data. The authors then study the ability of their proposed model to generalise five different skills to unseen object categories, showing superior performance compared to two considered baselines, and investigate the robustness of their policy to new environments and challenging textures.

**Summary Of Recommendation:**

Although the proposed modifications are simple and appear to work quite well for generalising to novel objects, the experiments section is unclear, making it difficult to access on what objects and how the proposed method was evaluated. And most importantly, it is difficult to assess the statistical significance of the presented results. Moreover, despite the authors acknowledging that "the robot may struggle to grasp novel objects with drastically different shapes or sizes than those seen in the training demonstration data, " this limitation could be at least partially addressed by incorporating depth data into the framework.

---

### Official Review · Reviewer_ZXLg · 2023-07-20

**Confidence:** 5
**Originality:** Very Good
**Technical Quality:** Excellent
**Clarity Of Presentation:** Excellent
**Impact:** 4

**Recommendation:**

Strong Accept: I recommend accepting the paper and will argue for my recommendation even if other reviewers hold a different opinion.

**Review:**

### Strengths:
1. Incorporating VLMs trained on captioned Internet data into robotics methods makes sense and is a valuable research direction, since more and more of this Internet data is being produced by millions of users, whereas robotics data is not growing at the same pace yet. The paper accomplishes this integration in an elegant way, providing this VLM information as an extra channel to the policy model but not overly restricting the policy.
2. Experiments with other user input modalities such as pointing are useful, since this is quite a practical way for humans to communicate with each other and with robots.
3. The design choices in the architecture are well-made to make the method immediately practical but also quite general in the long term. One small example: even though current instructions are focused on 1-2 objects, it is possible to add more objects to the mask by simply changing the pixel values in that channel, without needing to change the number of channels in the model. Another example: running the VLM only on the first frame, for computational scalability.
4. The experiments are thorough and constitute a strong investigation into different levels of generalisation, with especially impressive performance on new environments, which is promising for bringing robots into real situations.
5. The ablations are valuable and provide interesting insights, for example comparing the number of training objects vs number of training episodes and in which cases one matters more than the other.

### Weaknesses:

1. In the Introduction, the proposed method is framed in contrast to pipeline approaches, which are described as being more “brittle”. The reason given is that the latter control stages depend on accurate object detections in the earlier stages. This is intuitively true, but perhaps the paper could also benefit from experimental evidence for this claim, i.e. a direct experimental comparison against a pipeline approach to show that this new proposed method is less brittle. On Line 182, it is claimed that the MOO policy can be “robust to errors made by the VLM”, which is very interesting: some qualitative examples of this would be helpful to see and discuss.
2. The design decision of removing object names from the instruction after the objects are detected and before passing the instruction to the rest of the policy does have its benefits, because it forces the policy to learn to interpret the object detections in the mask channel. However, it also has its downsides. For example, it is not so obvious how the policy can learn to correct for mistakes of the VLM if the names of the objects it should interact with are removed from the instruction. Therefore, either an ablation on this design decision or a stronger justification for removing these names may be helpful here.
3. The idea of keeping only one-pixel detections from VLMs as inputs to the policy may have a slight tension with the argument made in the introduction. VLMs trained on static data currently have much more training data than robotics models, and so will likely do a better job of not only detecting the object centre but also detecting a good bounding box, and maybe even a segmentation. By keeping only a single pixel from the VLM output, the rest of this difficult work (i.e. finding the shape of the object) is shifted to the robotics policy, which has much less training data to learn from. Would it be possible to elaborate further on why discarding the bounding boxes from the VLM is beneficial here? Or if another model like Segment Anything were to be used as an additional channel, would the authors expect this to help or hurt performance? This is maybe more consistent with the argument made in the Introduction, which is that we want to shift the burden of open-set object detection away from the robotics policy and onto the web-scale VLMs as much as possible.

### Minor issues:

1. Minor referencing issue: should Table 1, when referenced on Line 302, be referenced as Appendix Table 1?

### Questions for rebuttal:

1. In Line 317, it is reported that the model cannot be made too large due to latency. For future work, would it make sense to have a larger high-level semantic reasoning model running at a lower frequency, and then also a smaller but higher-frequency low-level control model?
2. In the appendix, it is mentioned that prompts to OWL-ViT are prefixed with “An image of a”. This sounds reasonable, given that these are the captions that CLIP expects, and OWL-ViT uses CLIP. In the original CLIP paper, when performing zero-shot ImageNet classification, the authors report significant performance gains from using not just one but a set of many templates such as “A photo of”, “A blurry close-up of”, etc, and then aggregating scores across these templates for improved robustness. Would the authors consider trying out this technique for this robotics data? It may help with robustness, and lessen the need for the prompt re-mapping listed in the appendix.
3. How would this work when there are multiple apples in the scene, and the instruction is something like “move the apples next to each other”? In theory, OWL-ViT should give multiple good detections, and the mask channel should contain the 1.0 pixel in multiple places. Would the policy work in these cases, when the output of OWL-ViT has multiple instances of the same object category?



**Quality Of The Limitations Section:**

Limitations are addressed clearly

**Questions For Rebuttal:**

Please see the main review for questions for rebuttal.

**Robotics Focus:**

Sufficient demonstration on hardware

**Summary Of Paper:**

The paper proposes to integrate an object detection VLM into robotics policies by creating an extra channel for object detections, thus allowing the policy to generalise to completely unseen objects at test time. This approach is validated through large-scale real world experiments and a set of ablations. Its generalization capabilities are tested not only for novel objects, but also for novel environment conditions.

**Summary Of Recommendation:**

This is a strong paper which explores a very promising direction: integrating web-scale VLMs into robotics policies. The paper achieves this in an elegant way by giving the VLM output as an extra channel of input to the policy, without overly restricting the policy. The experiments are thorough, with impressive results. The paper could perhaps be improved a little further by strengthening the arguments in favour of some of the specific design decisions, but overall the method is sound and this is a valuable contribution to the robot learning community.

---

### Official Review · Reviewer_6xeG · 2023-07-22

**Confidence:** 4
**Originality:** Fair
**Technical Quality:** Good
**Clarity Of Presentation:** Very Good
**Impact:** 3

**Recommendation:**

Weak Accept: I recommend accepting the paper, but will not argue for my recommendation if the majority of other reviewers have a different opinion.

**Review:**

The paper is well-structured and presented. Main points are communicated well, with supporting visualisations which aid the reader greatly.

Strengths:
* The presented method is easy-to-understand and, as the authors mention quite easy to incorporate with any downstream policy architecture or upstream vision-language model (the 1-pixel segmentation masks concatenated to the input images channel-wise).
* extensive evaluation + the method is shown to work on a physical robot.

Weaknesses:
* the actual technical contribution is rather small and incremental - both the pre-trained VLM model and the policy architecture are taken off-the-shelf. What is changed is the input to the architecture with the 1x1 pixel masks (which have their drawbacks as discussed in the paper)
* the method depends on bboxes for the coordinates - what if the center of a bbox does not "fall on" the object - think of a crecend-shaped object where the center if its bbox is not on it. This is partly addressed in the limitations but would have been good to see how robust is the method to "moving" the 1x1 mask around - e.g. by simply adding noise to the actual coordinates. How much would that hurt performance? Is it sufficient for the 1x1 mask to be close to the object, assuming no clutter, or it has to be *on* it?

**Quality Of The Limitations Section:**

Limitations are addressed clearly

**Questions For Rebuttal:**

* lines 227-228 - we train on 106 objects and we pick 49 out of them to be the _seen_ ones, is that the case? If yes, why 49, how were they chosen? Also, in Figure 4, left, 49 are the seen objects only for Pick, correct, for the other skills it is still 16? Or it was 16 only during training, as no demonstrations with novel objects were collected for skills apart from pick.
* why do we need a template for the language? Doesn't the VLM model support more complex sentences?
* the data collection for the policy training is not discussed in the paper anywhere
* Figure 5 mentions Open-world objects. Can you elaborate what are those and how are they different from the unseen ones?

**Robotics Focus:**

Sufficient demonstration on hardware

**Summary Of Paper:**

The authors present MOO - a method that harnesses a pre-trained vision-language model that informs a transformer-based robot policy and allows it to better generalise to manipulating novel objects never seen during training. They present extensive evaluation of the method including variations of it which go beyond language-based inputs.

**Summary Of Recommendation:**

I recommend weak accept, happy to change to accept post-rebuttal, depending on discussion and answers.

---

### Official Review · Reviewer_VUMx · 2023-07-23

**Confidence:** 5
**Originality:** Good
**Technical Quality:** Very Good
**Clarity Of Presentation:** Good
**Impact:** 3

**Recommendation:**

Weak Accept: I recommend accepting the paper, but will not argue for my recommendation if the majority of other reviewers have a different opinion.

**Review:**

This research is a valuable combination of state-of-the-art VLM models and robotics control.
It is exciting to deal with open-world objects and open vocabulary in the age of LLM.
However, the fact that LLM can be used to support open vocabulary has probably already been demonstrated with RT-1 and others.
Therefore, the use of LLM to deal with open vocabulary is not a big surprise.

The quality of the paper and research is high.
However, the content is compressed and lacks technical details.

This paper appears to be an exhaustive evaluation with a wealth of experiments and evaluations.
The targeted tasks, data sets, instructions, and situations do not differ significantly from RT-1. It can be said that this is an incremental study.

The main text of this paper is based on the Appendix for details.
You will not know some of the details unless you read the Appendix.
I recommend that if they are mentioned in the main text, they be included in the main text, not the Appendix.

The paper seems to lack attention to detail.
For example, the citation format is disjointed and not uniform. Some information does not seem to be correctly described.
Authors are expected to review the descriptions for all citations.
In addition, there are too many ArXiv preprints for citations. (Almost half!) Preprints are not peer-reviewed, which is also a concern for the credibility of the paper. Therefore, preprint citations should be avoided whenever possible.

In the Introduction, I have the impression that the description of pre-trained VLMs trained on vision and language data (not robot observation data) and the description of learning and manipulation of robot policies are mixed up.
Each has been studied separately.
It is important that these separate things are integrated in this study.
However, it may be a good idea first to introduce a discussion of the earlier individual research areas in isolation.



**Quality Of The Limitations Section:**

Limitations are addressed clearly

**Questions For Rebuttal:**

There are several questions about the architecture of the proposed method.
One is why such a structure was necessary.
The proposed method seems to be a combination of several architectures, including Owl-ViT, FiLM, and Transformer.
However, the description of why this coupling method or combination was chosen is unclear.
Are these components substitutable in other similar models? Or is there a fundamental structure or function hidden somewhere?
There is no detailed description of the above, and it remains unclear.


What is the definition of the term Open-World Object Manipulation?
I couldn't find a description of the definition. It is unclear in what sense it is Open-World.

**Robotics Focus:**

Sufficient demonstration on hardware

**Summary Of Paper:**

This paper proposed an approach to robot mobile manipulation that leverages rich semantic knowledge captured by large-scale and visual language models.
The proposed method, called Manipulation of Open-World Objects (MOO), is an approach that obtains robot policies from human linguistic commands.
In addition, it showed higher performance for novel objects and novel instructions compared to existing state-of-the-art models such as RT-1.


**Summary Of Recommendation:**

The recommendation is based on the excitement surrounding open-world objects and open vocabulary in the era of LLM, along with the high quality of the paper and research. However, it is suggested to make some improvements in additional writing, argument organization, and formatting.

---

### Author Response · Authors · 2023-08-11
**Response to all reviewers (1/2)**

We thank the reviewers for their valuable feedback. We appreciate that Reviewer VUMx finds that the "quality of the paper and research is high", Reviewer 6xeG believes our method is "easy to incorporate with any downstream policy architecture or upstream vision-language model", Reviewer ZXLg thinks our work is a "strong paper which explores a very promising direction" with an "elegant way" of utilizing VLMs, and Reviewer VQYg finds the method "quite appealing and intuitive and is the right step in the direction of developing/learning general skills".

We have incorporated the helpful suggestions by running two additional experiments and improving the manuscript (updated manuscript edits are in red). Our main updates are as follows:

- A real-world evaluation of MOO's sensitivity to increasing amounts of noise in the single-pixel mask.
- An offline study of how MOO can operate with freeform non-templated instructions via an LLM interpreter
- Clarified the experimental setup description of "Unseen Objects" by disambiguating between "Unseen Objects from Seen Categories" and "Unseen Objects from Unseen Categories"
- Updated Figure 4 according to this more detailed breakdown
- Added a comprehensive list of all training and evaluation objects in the Appendix
- Added details in the Appendix on the data collection procedure for the new dataset we collected
- Updated the quality and formatting of citations

In addition, we would like to highlight the following common themes and our clarifications from the constructive feedback from reviewers:

1) Limited technical contribution

We wish to emphasize that the relative simplicity of the proposed method is by design in order to make minimal assumptions about the problem domain (e.g. calibrated cameras). Our proposed method is general and non-invasive, yet enables significant capabilities in practical robot learning settings in realistic situations with novel objects and backgrounds. We hope that the simplicity of our method combined with our strong empirical evaluations in the real world will motivate future works to consider incorporating similar object-centric conditioning.

2) Clarifying the notion of "Unseen Objects"

We recognize that the current presentation of how we organized our evaluation on "Unseen Objects" (in Section 4.1, Figure 3, Figure 4, Figure 9) can be improved. To do so, we break apart "Unseen Objects" into two types: a) "Unseen Objects from Seen Categories” and b) unseen objects of “Unseen Categories”. We consider an "Object Category" to be a class of objects sharing similarities in visual appearance, geometric, affordances, and/or requiring a particular type of grasping strategy, roughly following prior practical definitions [6]. For the evaluations in Section 4.1, our 47 "Unseen Objects" are 22 objects which are instances of "Seen Categories" (such as soda cans and chip bags) while 25 objects are instances of "Unseen Categories" (such as sunglasses or reflective metallic objects). We have updated Figure 4 in the manuscript by clearly separating results for "Seen Objects", "Unseen Objects, Seen Categories" and "Unseen Objects, Unseen Categories".

We highlight that our evaluation was explicitly designed to purposely evaluate new object categories that were significantly different from the training dataset, including difficult objects such as transparent bottles, miniature dinosaurs, or shiny coffee scoops. We hope that the new presentation of the results is more clear and better conveys our motivation and results. We expanded upon this in our response to Reviewer VQYg, and also updated our manuscript to clarify this evaluation design.

---

> ### Author Response · Authors · 2023-08-11
> **Response to all reviewers (2/2)**
>
> 3) Sensitivity to noise for single-pixel mask
>
> We originally noted the qualitative observation that MOO seemed to exhibit robustness to imperfect object-centric mask localization. We ran an additional experiment of evaluating a MOO policy with masks that contain artificially added localization noise. We study four cases where we ablate the mask while keeping the starting scene the same: (1) the baseline when the mask is at the centroid of the object, (2) when the mask is still on the object of interest but not at the centroid, (3) when the mask is off of the object entirely but still within roughly 5cm, and (4) when the mask is far and more than 5cm from the object of interest. We add the artificial noise manually by starting with the centroid mask and then manually ablating the masks. We run a total of 20 trials across 5 tasks involving one seen object (green rolling pin), one unseen object in a seen category (cold brew can), and three unseen object categories (egg plant, shiny sunglasses, transparent bottle). We find that performance degrades as more noise is added: case (1) achieves 5/5 successes, case (2) achieves 4/5 successes, case (3) achieves 3/5 successes, and case (4) achieves 3/5 successes. Qualitatively, we observe that the policy sometimes initially reaches for the inaccurate mask location, and is sometimes able to recover and re-approach the correct target object. In one notable example, the policy faced an off-center mask on the lower part of the clear bottle (a visually challenging object, since no transparent objects were in the training set) and grasped the clear bottle in the upper section, which is the correct strategy for re-orienting the bottle upright. Additionally, a few examples showed that the policy was able to retry after failures caused by inaccurate masks; this suggest that the policy does not just memorize going to the location of the mask, but instead also pays attention to semantics. We provide the table of quantitative results below:
>
> | Mask ablation/Task        | pick green rolling pin | pick eggplant | pick shiny glasses | place clear bottle upright | knock cold brew can over |
> |---------------------------|:----------------------:|:-------------:|:------------------:|:--------------------------:|:------------------------:|
> |   baseline centroid mask  |            1           |       1       |          1         |              1             |             1            |
> | off-center on-object mask |            0           |       1       |          1         |              1             |             1            |
> |    off-object <5cm mask   |            1           |       1       |          1         |              0             |             0            |
> |    off-object >5cm mask   |            1           |       0       |          1         |              0             |             1            |
>
>  In addition, we provide videos of qualitative rollouts and visualizations of the artificially ablated masks in the attachments in the individual reviewer responses.
>
>
> 4) Clarifying the usage of "Open-World"
>
> Our goal for focusing on our work as an "Open-World" robot learning system follows the definition of "Open-World" as introduced in prior robotics works in outdoor navigation [1], object navigation [2], and manipulation [3]; furthermore, we draw inspiration from computer vision with the original introduction of "Open-World" object detection challenges [4, 5].
>
>
> [1]: D. Shah et al., "ViNG: Learning Open-World Navigation with Visual Goals," ICRA 2021
>
> [2]: A. Majumdar et al., "Zson: Zero-shot object-goal navigation using multimodal goal embeddings," NeurIPS 2022
>
> [3]: Y. Mo et al., "Towards Open-World Interactive Disambiguation for Robotic Grasping," ICRA 2023
>
> [4] K.J. Joseph et al., "Towards open world object detection," CVPR 2021
>
> [5] M. Minderer et al., "Simple open-vocabulary object detection." ECCV 2022
>
> [6] L. Manuelli et al. "kpam: Keypoint affordances for category-level robotic manipulation," ISRR 2019

---

### Author Response · Authors · 2023-08-14
**Additional feedback/questions are welcome!**

Dear reviewers,

Thanks again for your insightful reviews! Please let us know if you have additional questions or concerns regarding our work and we'd be happy to provide more discussion and details!

---

### Decision · Program_Chairs · 2023-08-30

**Decision:**

Accept (Poster)

**Comment:**

The paper addresses the problem of open-world object manipulation using pre-trained vision language models. The architecture leveraged Owl-VIT for providing grounded object manipulation targets from language and a scene image. The extracted object masks are then provided to the RT-1 policy architecture. Experiments are demonstrated on an indoor mobile manipulation system with novel unseen objects.

The authors are requested to take into consideration the detailed feedback provided by reviewers. Specifically, (i) improving the technical details of the system (ii) commenting on to what extend the results are tied to the specific VLM and policy architecture used and (iii) robustness to perceptual errors/noise.